# Timescales of variation in diversity and production of bacterioplankton assemblages in the Lower Mississippi River

**Jason T. Payne, Colin R. Jackson, Justin J. Millar**[ID][¤]**, Clifford A. Ochs**[ID]*

Department of Biology, University of Mississippi, University, Mississippi, United States of America

¤ Current address: Malaria Atlas Project, University of Oxford, Oxford, England
* byochs@olemiss.edu

**Data Availability Statement:** Microbial sequence data are available in the NCBI SRA database under the BioProject ID PRJNA358603. Data for environmental parameters and bacterial production

## Abstract

Rivers are characterized by rapid and continuous one-way directional fluxes of flowing, aqueous habitat, chemicals, suspended particles, and resident plankton. Therefore, at any particular location in such systems there is the potential for continuous, and possibly abrupt, changes in diversity and metabolic activities of suspended biota. As microorganisms are the principal catalysts of organic matter degradation and nutrient cycling in rivers, examination of their assemblage dynamics is fundamental to understanding system-level biogeochemical patterns and processes. However, there is little known of the dynamics of microbial assemblage composition or production of large rivers along a time interval gradient. We quantified variation in alpha and beta diversity and production of particle-associated and free-living bacterioplankton assemblages collected at a single site on the Lower Mississippi River (LMR), the final segment of the largest river system in North America. Samples were collected at timescales ranging from days to weeks to months up to a year. For both alpha and beta diversity, there were similar patterns of temporal variation in particle-associated and free-living assemblages. Alpha diversity, while always higher on particles, varied as much at a daily as at a monthly timescale. Beta diversity, in contrast, gradually increased with time interval of sampling, peaking between samples collected 180 days apart, before gradually declining between samples collected up to one year apart. The primary environmental driver of the temporal pattern in beta diversity was temperature, followed by dissolved nitrogen and chlorophyll *a* concentrations. Particle-associated bacterial production corresponded strongly to temperature, while free-living production was much lower and constant over time. We conclude that particle-associated and free-living bacterioplankton assemblages of the LMR vary in richness, composition, and production at distinct timescales in response to differing sets of environmental factors. This is the first temporal longitudinal study of microbial assemblage structure and dynamics in the LMR.

used in creating Table 1 and Figs 1, 2, 3, and 8 are available at the GitHub website (https://github.com/jtaylorpayne/Timescales-project).

**Funding:** Funding was provided by the National Science Foundation Division of Environmental Biology grant 1049911 to Clifford Ochs and Colin Jackson.

**Competing interests:** The authors have declared that no competing interests exist.

## Introduction

In small streams, because of frequent and pronounced environmental disturbances in physical and chemical conditions, variation in microbial assemblage structure may be unrelated to timescale so that assemblages sampled closer in time may be as dissimilar as those sampled months apart [1]. In less stochastically disturbed aquatic systems, however, microbial assemblages appear to vary more predictably, and over the same temporal scales in which there is variation in diversity and/or activity of annual plant and animal assemblages [2]. For example, seasonally recurrent bacterioplankton assemblages have been observed in temperate marine environments [3, 4], lakes [5, 6], and even large rivers [7–10] associated with variation in day length, water temperature, hydrology, and nutrient concentrations.

Large river ecosystems of temperate zones are characterized by substantial temporal variation in nutrient and suspended sediment loads that is governed by their individual hydrographical underpinnings [11, 12]. At any given site within these systems, environmental fluctuation may be abrupt and unpredictable over brief periods of time responding to local storm events, or relatively gradual and deterministic due to climatic changes in temperature and/or precipitation within and among regional watersheds. Temporal dynamics of bacterial communities have been well described for many aquatic ecosystems, yet temporal variability in bacterioplankton assemblages of large rivers remains understudied. This is a significant gap in our knowledge of large river ecology, because of the importance of large rivers as conduits of nutrients to the sea [13]; because, as in other environments, bacteria are the most versatile and presumably the most important catalysts of biogeochemical transformations [14]; and because bacteria can reproduce rapidly and their community composition respond to environmental changes on a short-term basis [15].

From previous studies of the Mississippi River network, a system of multiple linked large rivers, we observed consistent and pronounced spatial variation in bacterioplankton assemblages. At a microhabitat level, assemblages attached to suspended particles (i.e. particle-associated bacterioplankton) were richer in bacterial operational taxonomic units (OTUs), and distinct in composition compared to free-living bacterioplankton [16, 17]. At a regional level, assemblages in major tributaries of the Mississippi River—the Illinois, Missouri, and Ohio rivers—were distinct in composition, presumably due to selection by particular environmental conditions of each river [16, 17]. Within the Mississippi River itself, planktonic microbial assemblages flowing downstream exhibited relatively large shifts in diversity after mixing at major confluences, while varying more gradually with increasing distance from confluences [17]. Clearly, as for other aquatic ecosystems, environmental selection processes structure

**Table 1. Coefficient of variation (%) in environmental variables at daily, weekly, and monthly timescales.**

| Variable | Daily (n = 8) | Weekly (n = 7) | Monthly (n = 12) |
|---|---|---|---|
| Temp | 4 | 9 | 55 |
| TSS | 14 | 41 | 64 |
| Chla | 27 | 33 | 44 |
| DOC | 6 | 7 | 13 |
| TDN | 2 | 11 | 51 |
| TDP | 8 | 23 | 23 |
| Discharge | 12 | 20 | 60 |

Abbreviations: Temp, water temperature; TSS, total suspended solids; Chla, chlorophyll *a*; DOC, total dissolved organic carbon; TDN, total dissolved nitrogen; TDP, total dissolved phosphorus.
n represents the number of dates per sampling interval.

bacterioplankton assemblages of this river network. However, in what taxonomic groups, of what magnitude, over what temporal scales, and in response to exactly what factors do assemblage changes occur? For instance, if one were to sample continuously over time at a single location in a large river water-column, in what respects and in concert with what environmental conditions, would the microbial plankton community vary? These questions address the relative importance to microbial community diversity and activity of stochastic variation over short time periods compared to over longer timeframes, in the context of an ecosystem marked by continuous, directional fluxes of water, chemicals, suspended materials, and microorganisms.

To address these questions, we documented variation in alpha diversity (within-sample richness of OTUs) and beta diversity (between-sample differences in composition) within and between particle-associated and free-living bacterioplankton assemblages over a range of temporal scales at a single site in the main channel of the Lower Mississippi River, the final segment of the largest river system in North America. Assemblages were collected on a daily and weekly basis in summer, and monthly over a year. Additionally, on each sampling date, we measured bacterial production and environmental variables. From these measurements, we determined the relationship of timescale to variation in assemblage diversity and production, and identified the strongest environmental correlates of variation. We hypothesized that bacterioplankton diversity and production of the LMR would vary less over shorter timescales and more over longer timescales, in relationship to gradual change in factors such as temperature, suspended sediments, algal biomass, and nutrient concentrations.

## Materials and methods

### Sample site and water collection

The Lower Mississippi River (LMR) was sampled on 23 dates between February 2013 and January 2014 (Fig 1A), near mid-channel directly off Mhoon Landing (34˚44'35.59" N 90˚26'58.03" W), near Tunica, Mississippi, USA (Fig 1B). Mhoon Landing is 76 river kilometers (rkm) below Memphis, Tennessee, and 426 rkm below Cairo, Illinois, where the Ohio River joins the Mississippi River, forming the LMR. At the Mhoon Landing sampling location the river is turbulent and deep (>7 m) with little evidence of vertical stratification in dissolved chemistry [18], and discharge generally ranges from roughly 7,000 to 27,000 $m^3$ $s^{-1}$ [19] depending on time of year (Fig 1A).

Sampling spanned three temporal scales (Fig 1A). Samples were collected once monthly, near the beginning of each calendar month, from 2 February 2013 to 11 January 2014, for a total of 12 monthly samples. At a finer scale, samples were collected weekly from 3 June to 15 July 2013, for a total of seven weekly samples. Finally, samples were collected daily from 24 June to 1 July 2013, for a total of eight daily samples. We chose to sample frequently during summer because this is a period of high bacterial production [18], and potentially a period in which a high degree of short-term temporal variation could be detected. On each date, sampling occurred between 10:00 and 13:00 h, and water was collected from mid-river at a depth of 0.5 m. Sterilized 1-L Nalgene sample bottles (n = 3) were used to collect water for chemical analyses and heterotrophic bacterial production, and sterilized 500-mL Nalgene sample bottles (n = 3) were used to collect water to analyze bacterioplankton assemblage structure. All bottles were stored in coolers containing river water to maintain ambient temperature during transportation to the laboratory (0.5–1.5 h) for additional measurements, sample fractionation, and preservation.

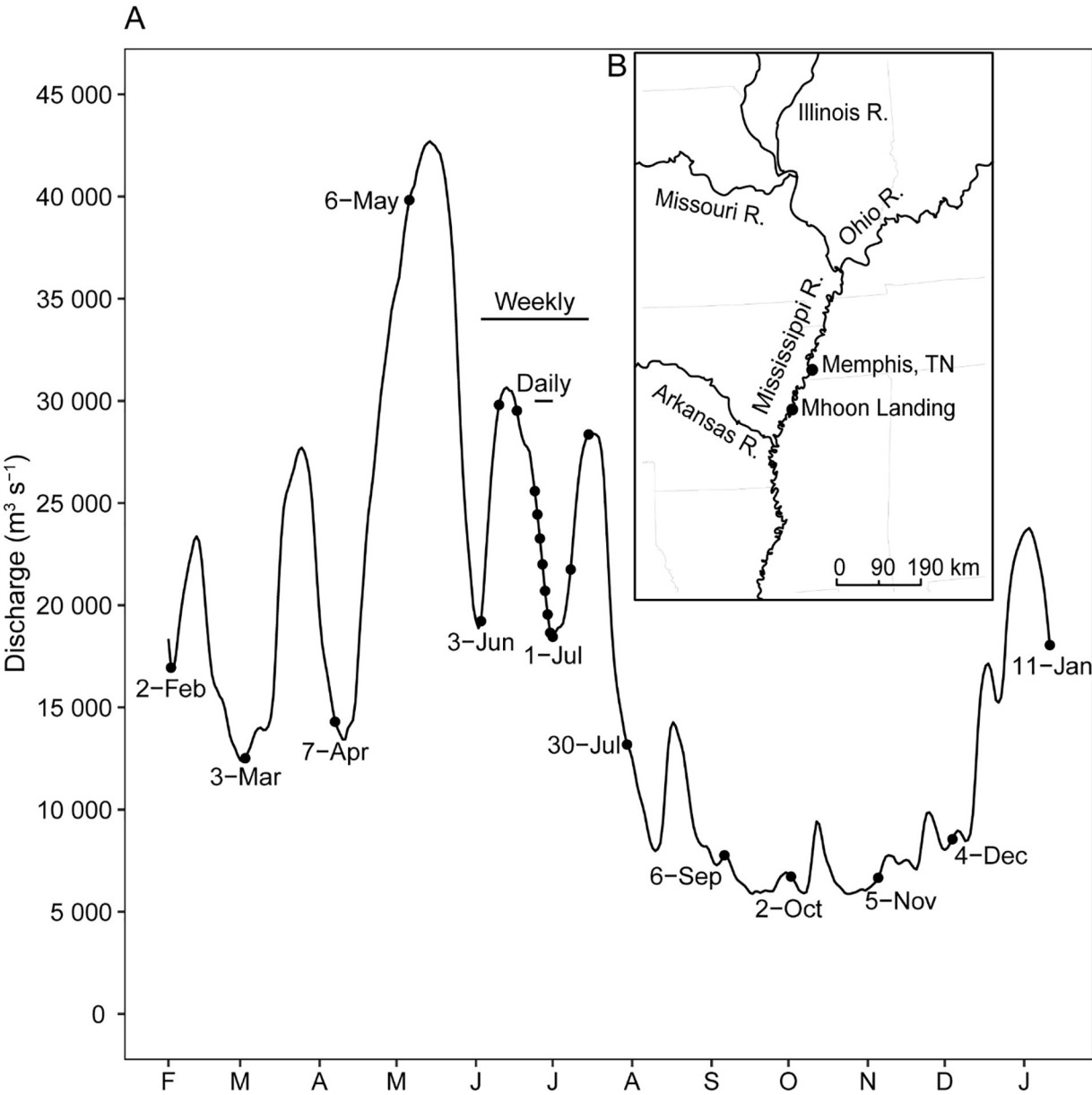

**Fig 1.** Hydrograph of discharge of the Lower Mississippi River at Mhoon Landing, Mississippi between February 2013 and January 2014 **(A).** Points on hydrograph represent sample dates. Monthly sample dates (n = 12) are labeled by date, while horizontal bars indicate weekly (3 June to 15 July 2013, n = 7) and daily sampling (24 June to 1 July 2013, n = 8) periods. Discharge measurements were calculated using gage height data collected daily by the U.S. Army Corps of Engineers at Helena, Arkansas located 40 rkm below Mhoon Landing. Map of a portion of the Mississippi River Basin indicating sample location (Mhoon Landing) relative to Memphis, Tennessee, and major river tributaries (B).

This field study did not involve endangered and protected species, and all samples used in this study were collected from a public river waterway for which permission to obtain samples was not required.

## Environmental measurements

Water temperature was measured in the field using a Hawkeye Digital Sonar H22PX-B. In the laboratory, sub-samples (100–200 mL) were filtered through ashed 47-mm diameter,

Whatman GF/F filters. For preservation, filters and filtrates were frozen at -60˚C or -20˚C, respectively. Samples remained frozen < 18 months prior to testing. Total suspended sediment (TSS) concentrations were measured gravimetrically on filters after drying at 60˚C. Chlorophyll *a* (Chla) concentrations were assayed by spectrophotometry of pigments extracted in 90% $NH_4OH$-buffered acetone for 24 h at 5˚C [20]. Total dissolved organic C (DOC) and total dissolved N (TDN) were measured in filtrates using a Shimadzu Total Organic Carbon Autoanalyzer, while total dissolved P (TDP) concentrations were assessed using standard spectrophotometric methods [20]. Units for these environmental measures pertinent to all analyses are given in Fig 2.

## DNA extraction and sequencing

From the 500-mL sample bottles, 100-mL subsamples were removed for serial filtration (<5 mm Hg vacuum). Subsamples were initially passed through sterile Millipore 3-μm pore-size polycarbonate filters, and the filtrate immediately filtered through sterile Millipore 0.22-μm pore-size polyethersulfone filters. Particles collected in the first filtration include particle-associated cells, cells, or colonies >3 μm in size (hereafter referred to as particle-associated cells). Particles collected in the second filtration step include smaller (0.22–3 μm) bacteria, assumed to be mostly free-living [16, 17]. Filters were stored at -20˚C before molecular processing. Samples remained frozen < 18 months prior to testing.

DNA was extracted from filters using PowerWater DNA isolation kits (MoBio, Carlsbad, California). The bacterial 16S rRNA gene was amplified and sequenced using methods modified from Kozich et al. [21], and described previously [17, 22]. Briefly, DNA was amplified using standard forward (5′ –GTGCCAGCMGCCGCGGTAA) and reverse (5′ –GGACTACHVG GGTWTCTAAT) primers adapted with dual-index barcodes for Illumina MiSeq next generation sequencing [21], and run through 30 cycles of denaturation (95˚C) for 20 s, annealing (55˚C) for 15 s, and elongation (72˚C) for 2 min, and a final elongation (72˚C) for 10 min. Negative (no template) controls were used in all amplifications and consistently gave negative results. Such negative amplifications were also used as blanks in sequencing, yielding no sequence data. Positive controls were not needed as we have used these procedures successfully for a variety of sample types [17, 23, 24]. PCR products were normalized by sample using Sequal-Prep Normalization Plates (Life Technologies, Grand Island, New York), pooled, and sequenced using an Illumina MiSeq platform located at the Molecular and Genomics Core Facility at the University of Mississippi Medical Center. All sequences can be accessed in the NCBI SRA database under the BioProject ID PRJNA358603.

## Sequence processing

Sequence data were processed using the bioinformatics software mothur [25] by a procedure modified from Payne et al. [17]. Briefly, the SILVA rRNA database (release 119) was used to align sequences with reference V4 sequences [26], and all unaligned sequences were discarded in addition to homopolymers >8 bp. Before classification, sequences differentiated by ≤2 bp were merged, and potential chimeras identified by UCHIME [27] removed. Sequences were classified using the RDP database (Release 11, September 2016) [28]. Non-bacterial lineages (e.g. Archaea, Eukarya, and mitochondria) were then removed. As RDP classification does not distinguish between cyanobacteria and chloroplast lineages at the phylum-level, chloroplast sequences were removed in a subsequent step (see below). Finally, all remaining sequences were clustered into OTUs based on ≥97% similarity.

Sequence data were processed further and analyzed in R version 3.5.1 [29]. OTU and taxonomy tables generated by *mothur* were imported into R and merged with environmental

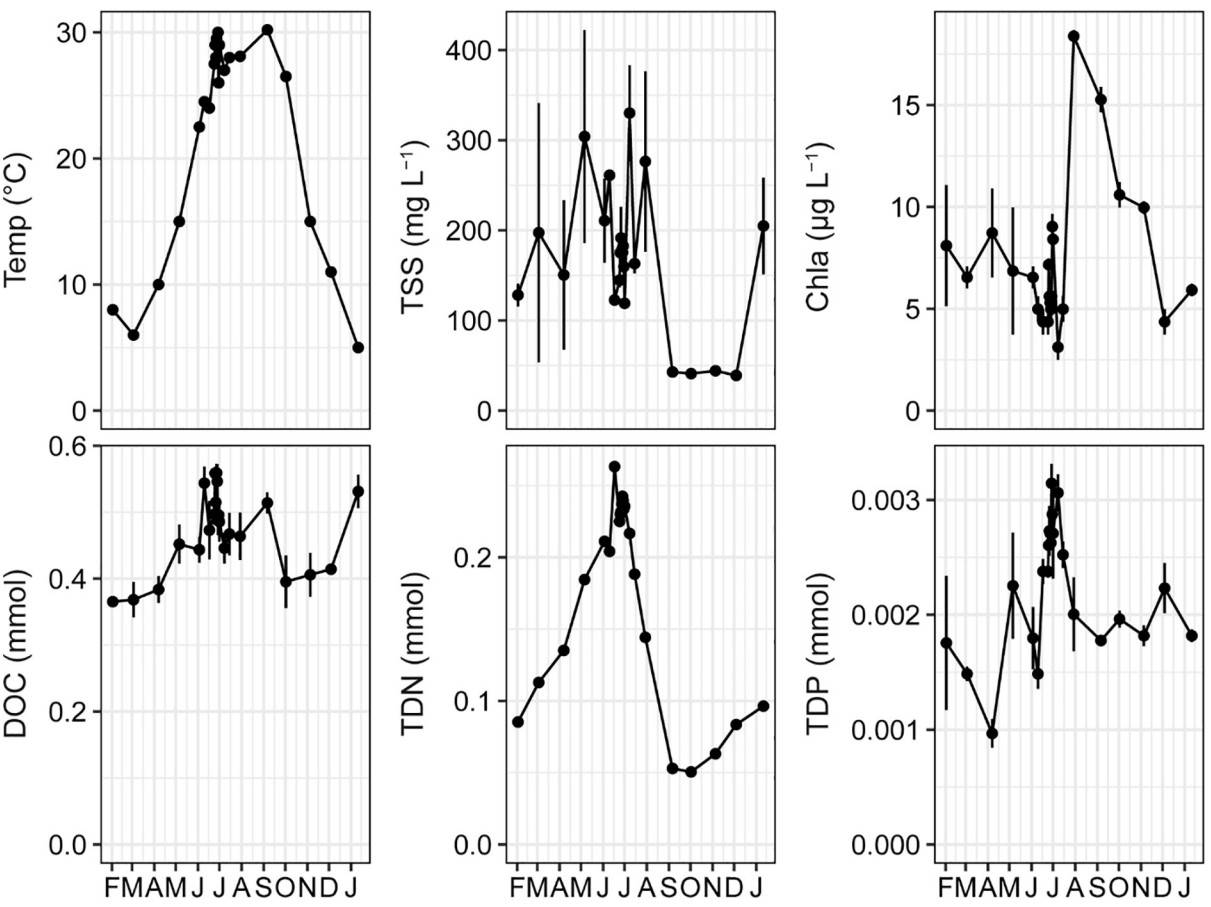

**Fig 2. Environmental variables measured in Lower Mississippi River water between February 2013 and January 2014.** Abbreviations: Temp, water temperature; TSS, total suspended solids; Chla, chlorophyll *a*; DOC, total dissolved organic carbon; TDN, total dissolved nitrogen; TDP, total dissolved phosphorus. Except for water temperature, parameter measurements are presented as means (± SE) for each date, n = 2–3. For clarity, sample dates are connected by lines. These lines are not intended to convey patterns of variation at shorter time intervals than what is shown.

metadata using the microbiome analysis software *phyloseq* version 1.14.0 [30]. OTUs identified as belonging to chloroplast lineages were removed from the dataset.

Alpha diversity (i.e. richness of bacterial OTUs within samples) was determined from an untrimmed dataset (i.e. containing singleton OTUs) using the phyloseq function "estimate_richness". Beta diversity (i.e. differences in assemblage composition) was evaluated after removal of OTUs with fewer than one read in 10% of the samples (i.e. potentially erroneous and rare OTUs) were removed from the dataset. OTU counts were then normalized using edgeR [31].

## Bacterial production measurements

Bacterial production was determined based on radiolabeled isotope incorporation. Leucine ($^3$H-leucine) and thymidine ($^3$H-thymidine) (Moravek Biochemicals) at specific activities of approximately 100 Ci mmol$^{-1}$ were used to determine synthesis rates of proteins and DNA, respectively [20]. Production of the total assemblage was measured using whole-water samples, while production of free-living cells was measured in sample water filtered through sterile 47-mm diameter, 3-µm pore-size Millipore polycarbonate filters [18].

Production measurements were made using a microcentrifuge procedure modified from Kirchman [32]. Triplicate bulk and filtered water samples (1.5 mL) were added to 2-mL microcentrifuge vials along with a saturating concentration of 60 nM $^3$H-leucine or $^3$H-thymidine [18]. A control tube for every treatment was prepared by adding trichoroacetic acid (TCA) immediately after isotope addition (see below). Thus, there were a total of 16 vials used per sample event. Incubations were initiated in the field beginning immediately after sample collection. Vials were incubated in river water at ambient temperature for 1 h, then placed on ice for 5 min, after which 94 μL of 80% TCA was added to halt isotope uptake. In the laboratory, vials were centrifuged at 18,000 rpm for 10 min, and the supernatant removed. Cold 5% TCA (1 mL) was then added to each vial followed by vortexing, centrifugation, and removal of supernatant. Finally, 1 mL of ice-cold 80% ethanol was added, followed by the washing steps above. Pellets were dried at room temperature overnight, and 1 mL of Fisher ScintiSafe Plus 50% scintillation fluid added to vials, followed by further vortexing. Radioassays were run on a Perkin-Elmer Tri-Carb 2810 TR liquid scintillation counter. Radioisotope-uptake calculations for $^3$H-leucine representing biomass production, and $^3$H-thymidine representing cell reproduction, were made as explained in Wetzel and Likens [20]. Production of all cells (whole-water) and free-living cells (<3-μm fraction) was determined directly, while production of particle-associated cells was determined by difference.

## Statistical analysis

Univariate statistics were performed using the package *car* [33], while multivariate statistics were performed using either *phyloseq* or *vegan* version 2.5–3 [34]. Graphics were generated using ggplot2 version 2.1.0 [35].

Levene's Test was used to detect homogeneity of variance in bacterial alpha diversity between particle-associated and free-living samples. Variance in assemblage alpha diversity, beta diversity, and production between samples collected over daily (24-Jun– 1-Jul, n = 8), weekly (3-Jun– 15-Jul, n = 7), and monthly (2-Feb– 11-Jan, n = 12) sampling intervals were shown using boxplots. Mood's median tests were used to compare the medians. Post-hoc tests were run using the function "pairwiseMedianTest" in the *rcompanion* package [36].

Beta diversity was quantified using Bray-Curtis dissimilarity matrices. To visualize whether bacterial samples collected closer in time were more similar in composition, mean pairwise dissimilarities were plotted against Euclidian distances in sample date. Differences in composition between particle-associated and free-living samples were also visualized using non-metric multidimensional scaling (NMDS) ordinations. Envfit (package *vegan*) analysis was then used to determine abundant bacterial OTUs that correlated with separation of samples in NMDS space.

Permutational multivariate analysis of variance (function "adonis" in the package *vegan*) was used to test for significant differences in beta diversity between groups of samples (e.g. between particle-associated and free-living, or between samples collected at daily, weekly, and monthly timescales) [37]. Permutated distance-based test for homogeneity of multivariate dispersion (function "PERMDISP2" in the package *vegan*) was then used to test for significant differences in the variance in beta diversity between sample groupings [38].

Environmental drivers of particle-associated and free-living beta diversity were determined by model selection using corrected Akaike information criterion (AIC$_c$) [39] in the software Plymouth Routines in Multivariate Ecological Research (PRIMER) 7.0 [40]. Environmental variables in models included: temperature, TSS, Chla, DOC, TDN, TDP, and discharge. Prior to AICc analysis, a cross-correlation matrix analysis of candidate predictors was performed. Predictors having a correlation coefficient $\geq$ 0.8 were not both included in the model for

community composition. Relative variable importance (RVI) scores were calculated for each environmental variable based on appearance in the $AIC_c$-best models, and a pseudo-$R^2$ was calculated for the best models to quantify their fit to the data. Variables that had $RVI > 0.5$ were considered most important.

Samples were also used to assess patterns of variation in relative abundances of bacterial OTUs. Plots were created in package *ggplot2* using the function "stat_smooth". Local polynomial regression fitting (function "loess" in the package *ggplot2*) was used to display patterns of variation in relative abundances. 95% confidence intervals were plotted around regression lines.

## Results

### Patterns in the river environment

Over the course of the study, water temperature ranged from 5˚C on 11-January to 30˚C on 29-June and 9-September (Fig 2). TSS concentrations peaked during high discharge on 6-May and 8-July, while Chla concentrations were at a maximum during low discharge on 5-November. TDN corresponded closely to the pattern in the river hydrograph ($r = 0.70$, $p = 0.01$). DOC ($r = 0.42$, $p = 0.26$) and TDP ($r = 0.33$, $p = 0.28$), in contrast, did not vary with discharge. Seasonal and annual variability of these variables in the LMR are tightly coupled with climatic and hydrologic conditions inherent to the river's large watershed, as documented previously [18, 41, 42].

To compare patterns in the timescales of variation, for each environmental variable we calculated the coefficient of variation (CV) for measurements taken over daily (24-Jun– 1-Jul, n = 8), weekly (3-Jun– 15-Jul, n = 7), and monthly (2-Feb– 11-Jan, n = 12) sampling intervals. For all variables, relative variation increased with timescale of measurement (Table 1).

### Patterns in bacterial alpha diversity

A total of 4,774,499 bacterial sequences were recovered from particle-associated and free-living bacterial fractions, corresponding to 43,289 bacterial OTUs. High-quality sequence reads for individual sample sets ranged between 1,136 and 457,102 sequences. On all dates, bacterial alpha diversity (i.e. richness of OTUs) was greater within particle-associated components compared to the free-living counterpart (range = 1.1 to 7.1 times), with peaks of richness for both fractions in mid-summer (Fig 3). However, the degree of variation in richness over the year was not significantly different between the different components of the microbial community (Levene's Test, $p = 0.264$).

There was no significant difference in median particle-attached richness (Mood's median tests: $p < 0.001$) among daily, weekly, and monthly timescales (Fig 4A). Furthermore, there was a similar degree of variability for richness of this fraction among all timescales. There was more variability in free-living richness at short time intervals (i.e. daily and weekly timescales) (Fig 4A), but there was no significant difference among richness medians.

### Patterns in bacterial beta diversity

In general, both particle-associated and free-living components were more similar in composition on daily and weekly timeframes than on a monthly timeframe. However, the pattern was not linear for either group. Instead, dissimilarity exhibited a roughly parabolic pattern (Fig 5). Assemblages became increasingly dissimilar in composition with separation in time up to six months, after which the trend was for a gradual decrease in dissimilarity. If we disregard year, these trends indicate that assemblages occurring closer in time, whatever the time of year, are increasingly alike in composition. Furthermore, this pattern of nonlinearity shows that the LMR microbiome varies along seasonal gradients.

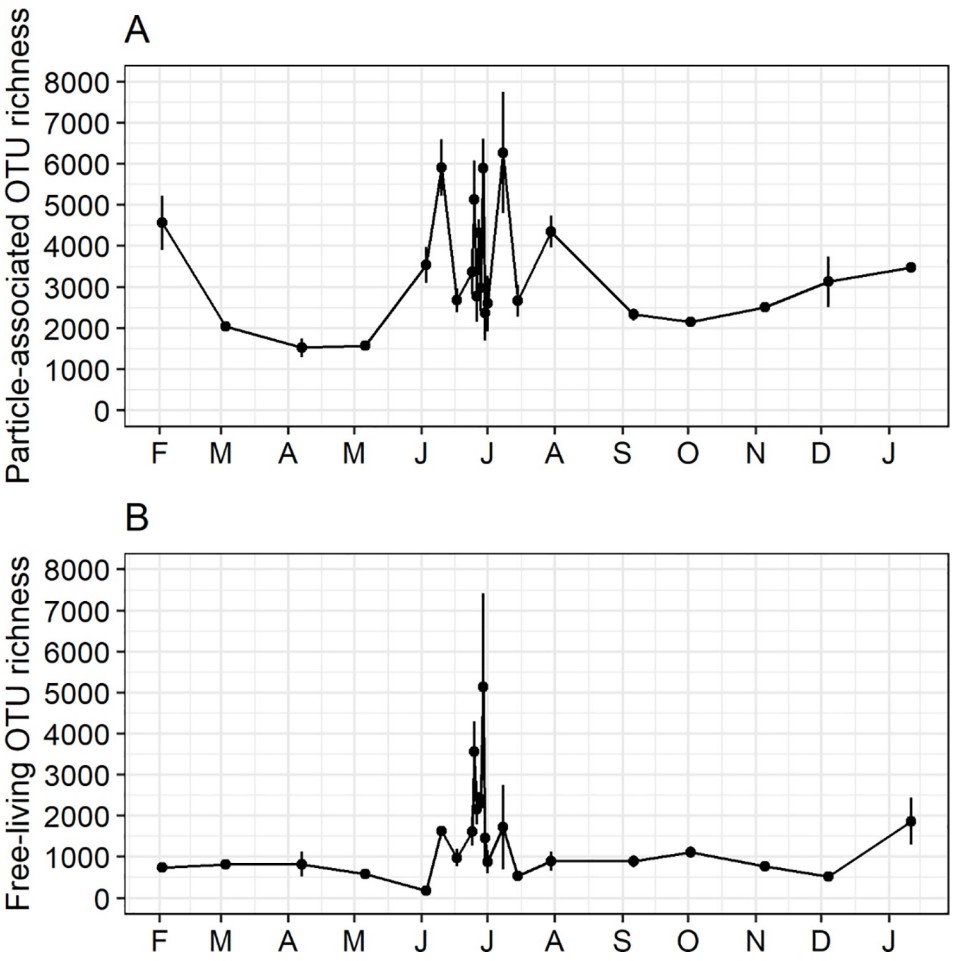

**Fig 3. Temporal patterns in bacterioplankton alpha diversity measured using richness of OTUs.** Differences in richness of OTUs in (A) particle-associated and (B) free-living bacterioplankton assemblages collected on 23 dates from February 2013 to January 2014.

While particle-associated and free-living assemblages were distinct in composition (adonis: $R^2 = 0.08$, $p < 0.001$), they were similarly variable in composition (PERMDISP2, $p = 0.172$). For both particle-associated and free-living components, there was less variability in beta diversity at a daily timescale compared to a weekly timescale, with the greatest variability occurring at a monthly timescale (Fig 4B). Furthermore, median beta diversity values increased significantly ($p < 0.001$) with the increase in sampling interval for both components.

The best models selected by AICc explained 49% and 38% of the variation in particle-associated and free-living assemblage composition, respectively (Table 2). The best model explaining variation in particle-associated beta diversity included water temperature as the primary factor (RVI = 0.93) and TDN was also important (RVI = 0.62). Water temperature was the main factor (RVI = 0.81) in the model explaining variation in free-living assemblages, followed by Chla (RVI = 0.53).

## Patterns in relative abundances of bacterial taxa

At a broad taxonomic level, particular bacterial phyla exhibited distinct patterns in their proportional abundance over the year (Fig 6). Proportions of Proteobacteria were fairly constant

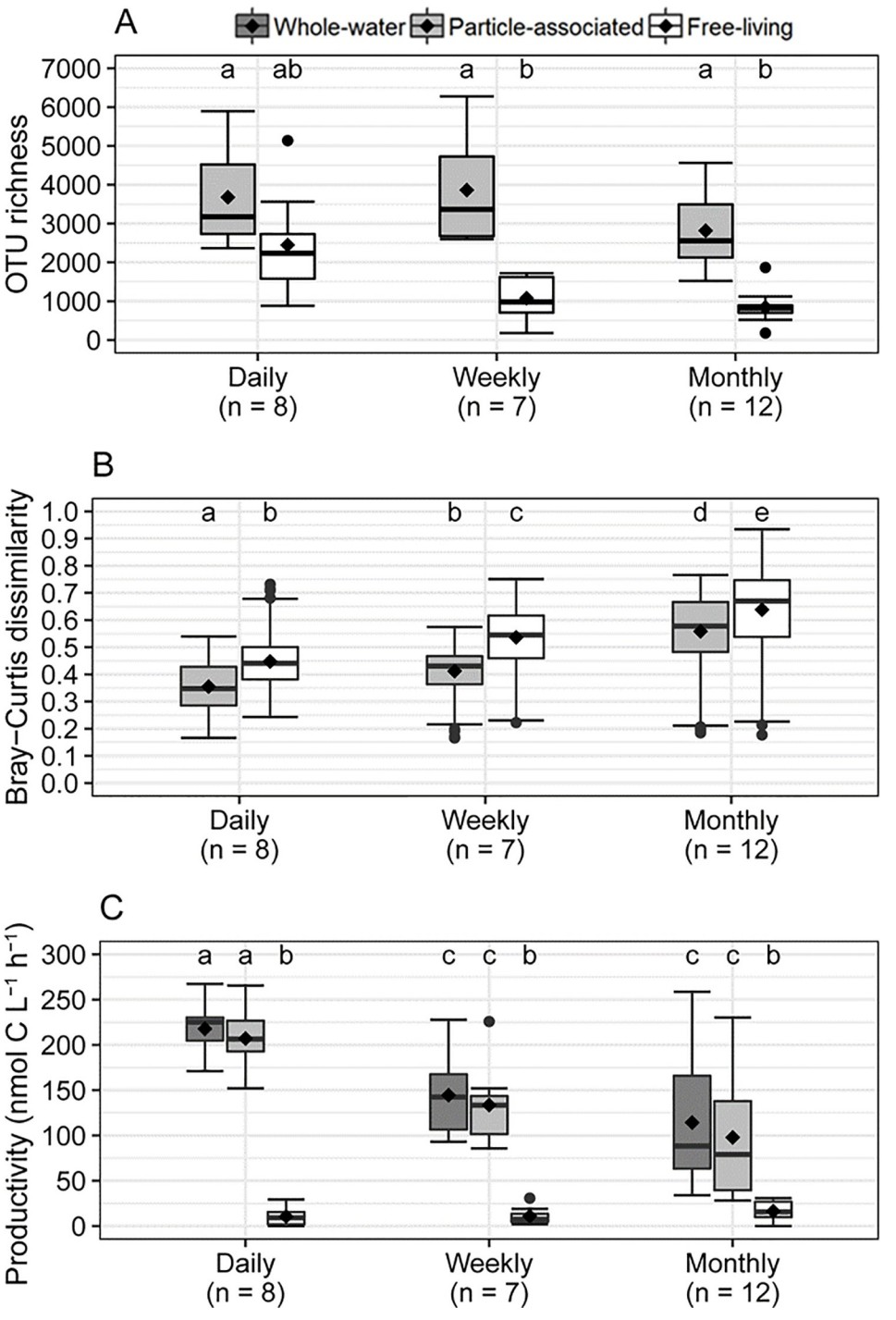

**Fig 4.** Boxplots showing variance in bacterial assemblage **(A)** OTU richness **(B)** Bray–Curtis dissimilarity, and **(C)** production ($^3$H-leucine) among sampling timescales. Boxes show medians (dark lines), averages (diamonds), and inter-quartile ranges. Whiskers indicate data within 3X inter-quartile ranges, and points are outliers. Letter(s) above boxes indicate the groups of samples that are significantly different in their medians (Mood's median tests: $p < 0.001$). Sample sizes are presented for each timescale.

over much of the sampling period, but trended upward from November to January in both particle-associated and free-living components. Relative abundances of other phyla, in

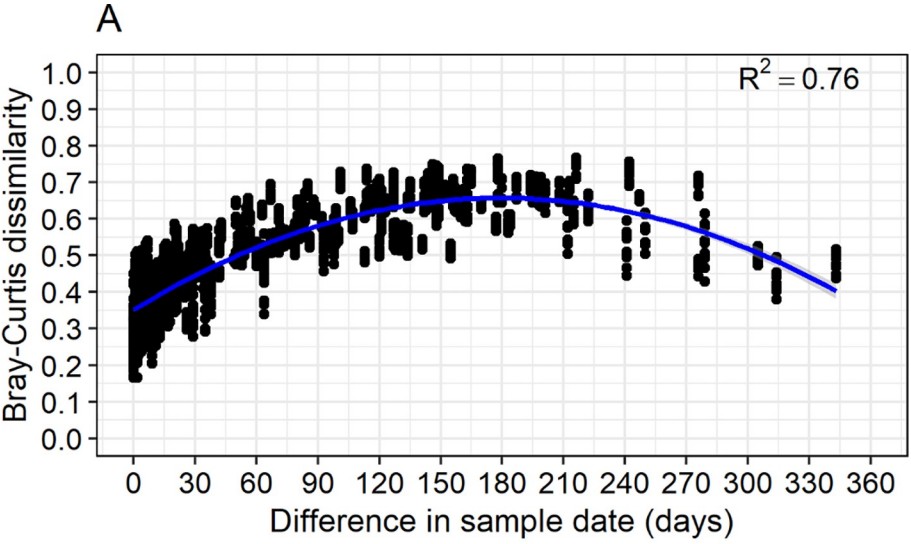

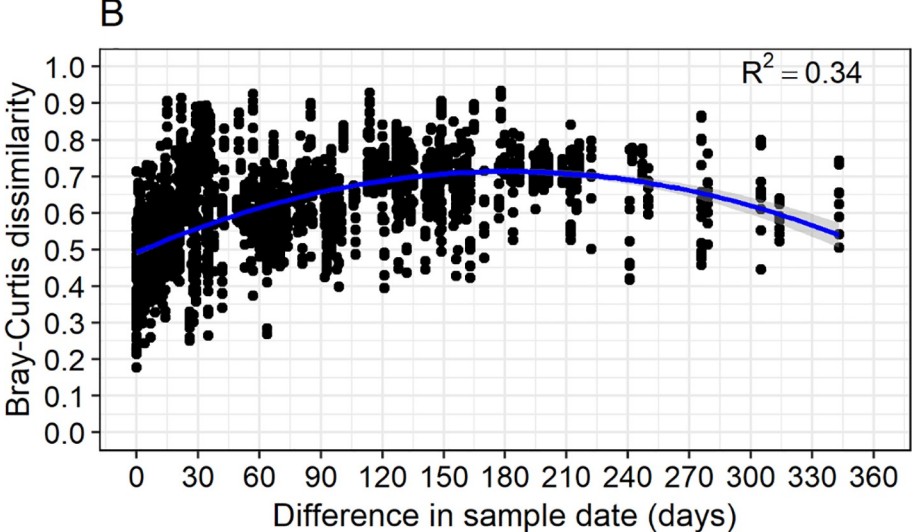

**Fig 5. Temporal patterns in bacterioplankton beta diversity measured using Bray-Curtis dissimilarity.**
Relationships between (A) particle-associated and (B) free-living dissimilarities and interval of time between sample
dates. Points represent pairwise dissimilarities calculated from bacterioplankton assemblages collected between 1 and
343 days apart.

contrast, were more closely related to seasonal changes in water temperature and/or the river
hydrograph. Sequences classified as Bacteroidetes and Verrucomicrobia were abundant in
assemblages collected in cooler water in spring and winter. Decreased proportions of these
taxa, in particular Bacteroidetes, in warm river conditions corresponded with increased pro-
portions of Acidobacteria in summer, and Planctomycetes throughout summer and fall. Cya-
nobacteria increased in proportion in late-summer and into early fall when the river was at a
minimum in discharge, TSS load, and turbidity. Proportions of Actinobacteria increased from
late-summer to winter, after which they strongly dominated free-living assemblages during the
period of least discharge from mid-July to December. However, members of this phylum were
much less abundant in particle-associated assemblages sampled during this time.

**Table 2. Summary of results of relationships between variation in environmental variables and bacterial assemblage beta diversity including relative importance of variables based on model selection using AICc (Akaike's Information Criterion corrected for small samples).**

| Analysis | Relative variable importance and sum of Akaike weights (sum *wi*) for each variable | Pseudo-$R^2$ of AICc-best model |
|---|---|---|
| Particle-associated | **Temperature** (sum *wi* = 0.93) > | 0.49 |
| | **TDN** (sum *wi* = 0.62) > | |
| | Discharge (sum *wi* = 0.43) > | |
| | TDP (sum *wi* = 0.42) > | |
| | Chla (sum *wi* = 0.38) > | |
| | DOC (sum *wi* = 0.31) > | |
| | TSS (sum *wi* = 0.28) | |
| Free-living | **Temperature** (sum *wi* = 0.81) > | 0.38 |
| | **Chla** (sum *wi* = 0.53) > | |
| | Discharge (sum *wi* = 0.45) > | |
| | TDP (sum *wi* = 0.41) > | |
| | TDN (sum *wi* = 0.40) > | |
| | DOC (sum *wi* = 0.40) > | |
| | TSS (sum *wi* = 0.33) | |

Variables in bold type had a sum of Akaike weight (sum *wi*) greater or equal to 0.5 and thus were considered relatively important.

A NMDS ordination confirmed these seasonal patterns of change in composition of particle-associated and free-living bacterioplankton assemblages (Fig 7). Particle-associated assemblages separated in time in a roughly clockwise pattern in NMDS space, from winter to spring to summer to fall, revealing changes in composition over time in a gradual manner. While a cyclical pattern was not apparent for the free-living fraction, the ordination shows that both particle-associated and free-living assemblages collected nearly a year apart trended towards increased similarity in composition.

Envfit analysis identified several OTUs that were correlated ($R^2 \geq 0.55$) with bacterioplankton assemblages collected in spring and winter (Fig 7; Table 3). These OTUs were related to Bacteroidetes (OTU41, OTU53, OTU60, and OTU66), Betaproteobacteria (OTU08 and OTU32), and Verrucomicrobia (OTU21). The associations between bacterial OTUs and assemblages collected in summer were weaker in comparison, however, free-living assemblages in late-summer and fall correlated with OTUs identified to the Actinobacteria order Actinomycetales (OTU02 and OTU04) and an unclassified member of Betaproteobacteria (OTU13).

## Patterns in bacterial production

Rates of whole-water bacterial production measured by the two radioisotopes were very similar, ranging over the year from about 30 to 300 nmol C L$^{-1}$ h$^{-1}$ (Fig 8). The temporal pattern correlated strongly with temperature, $R^2$ = 0.68 and 0.78 for [3]H-leucine and [3]H-thymidine incorporation, respectively (p < 0.001 for each), increasing from spring through late summer, and declining to minimum values in winter. Particle-associated production was usually much greater than for free-living cells. On average, attached bacteria represented 87.9% (standard error = 2.3%) of new biomass measured by [3]H-leucine uptake (protein synthesis), and 89.3% (standard error = 2.7%) measured by rates of [3]H-thymidine uptake (cell division) in whole-water.

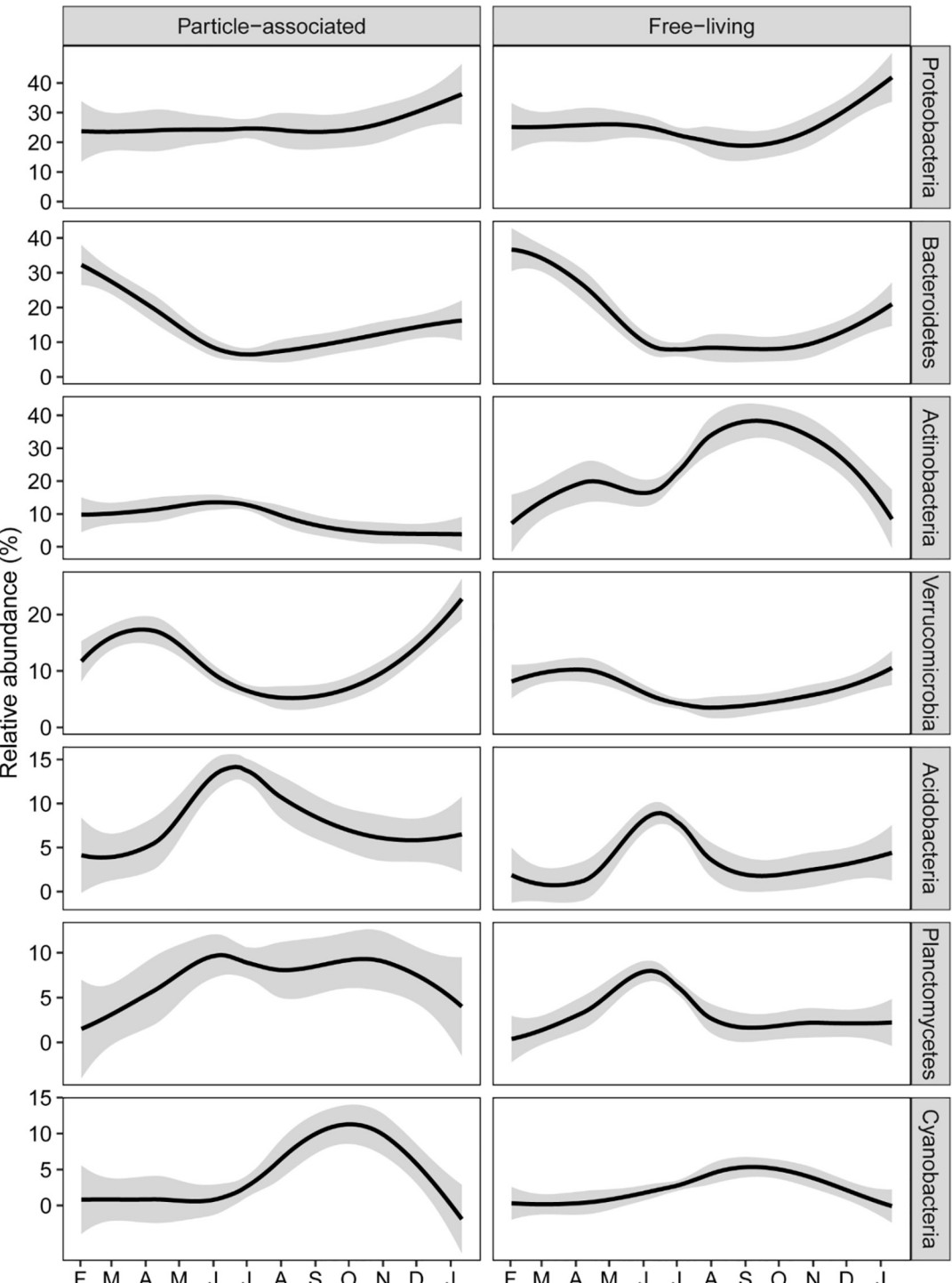

**Fig 6. Temporal patterns in relative abundances of bacterial phyla sequenced from particle-associated and free-living bacterioplankton assemblages.** Lines were made using local polynomial regression fitting (loess). Shading around lines indicate 95% confidence intervals.

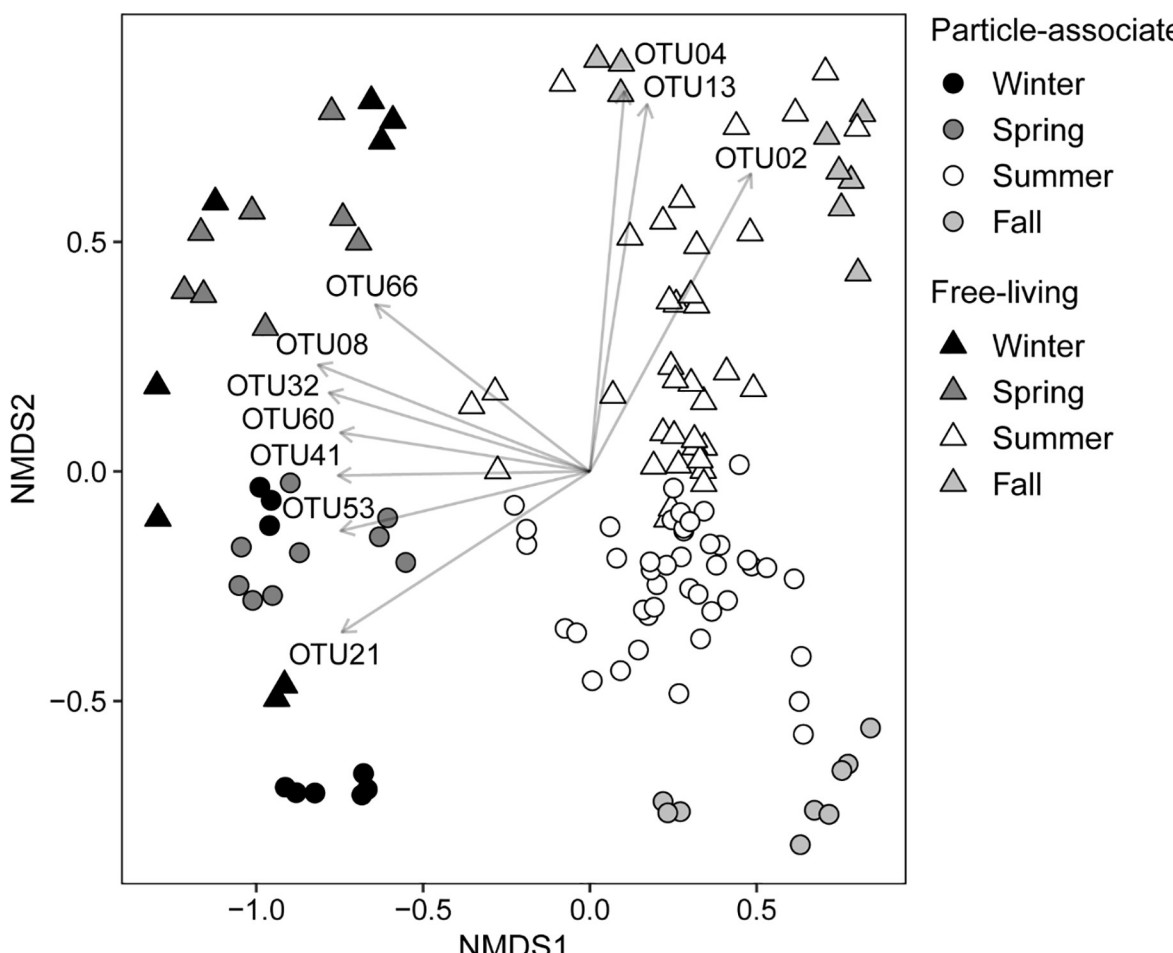

**Fig 7. A NMDS ordination showing seasonal changes in composition of particle-associated and free-living bacterioplankton assemblages.** Stress for the ordination equaled 0.11. Arrows indicate bacterial OTUs correlated (Envfit analysis: $R^2$ = 0.55–0.72, p = 0.001) with the ordination. Identifications of OTUs (RDP classification) are as follows: (OTU02 and OTU04) order Actinomycetales (Actinobacteria); (OTU08) family Comamonadaceae (Betaproteobacteria); (OTU13) class Betaproteobacteria (Proteobacteria); (OTU21) *Prosthecobacter* (Verrucomicrobia); (OTU32) *Methylophilus* (Proteobacteria); (OTU41 and OTU53) *Flavobacterium* (Bacteroidetes); (OTU060) phylum Bacteroidetes; and (OTU66) family Cytophagaceae (Bacteroidetes). Complete identifications of OTUs and specific $R^2$ values of correlations are presented in Table 3.

There was less variability in whole-water and particle-associated production at daily and weekly timescales compared to the monthly timescale, while there was a similar amount of variability in free-living production among all timescales (Fig 4C).

## Discussion

The physical environment and associated plankton communities of flowing waters are continuously in downstream flux. Hence, at a particular riverine location, plankton assemblages could diverge rapidly in diversity and metabolic activity in response to flow-mediated immigration and emigration. Adding to the potential for rapid change in community diversity with flow rate is the reproductive potential of resident biota. Having potentially high rates of turnover, while also subject to continuous downstream flux, the bacterioplankton microbiome of a particular river location potentially could vary as much on the order of days or weeks as among months or seasons. However, in contrast to low-order streams and rivers, the immense

**Table 3. OTUs that correlated (Envfit analysis, $R^2$) with bacterioplankton assemblages plotted in NMDS space.**

| OTU | Phylum | Class | Order | Family | Genus | $R^2$ |
|---|---|---|---|---|---|---|
| OTU02 | Actinobacteria | Actinobacteria | Actinomycetales | | | 0.65 |
| OTU04 | Actinobacteria | Actinobacteria | Actinomycetales | | | 0.70 |
| OTU08 | Proteobacteria | Betaproteobacteria | Burkholderiales | Comamonadaceae | | 0.72 |
| OTU13 | Proteobacteria | Betaproteobacteria | | | | 0.67 |
| OTU21 | Verrucomicrobia | Verrucomicrobiae | Verrucomicrobiales | Verrucomicrobiaceae | *Prosthecobacter* | 0.68 |
| OTU32 | Proteobacteria | Betaproteobacteria | Methylophilales | Methylophilaceae | *Methylophilus* | 0.64 |
| OTU41 | Bacteroidetes | Flavobacteriia | Flavobacteriales | Flavobacteriaceae | *Flavobacterium* | 0.57 |
| OTU53 | Bacteroidetes | Flavobacteriia | Flavobacteriales | Flavobacteriaceae | *Flavobacterium* | 0.58 |
| OTU60 | Bacteroidetes | | | | | 0.57 |
| OTU66 | Bacteroidetes | Cytophagia | Cytophagales | Cytophagaceae | | 0.55 |

OTUs were classified using the RDP database (release 11, September 2016).

volume of large rivers may buffer these systems from rapid environmental or biological variation. In that case, we would expect microbiome assemblage structure and function to vary slowly, following seasonal or annual patterns in regional environmental drivers, rather than transiently-acting factors associated with random local disturbances. We documented temporal patterns of variability in bacterioplankton microbiome structure and production at a single location on the LMR over a range in timescales, from days up to a year. Our time-nested sampling design and results allow us to assess the extent to which constant habitat turnover and environmental variation drives community change.

Differences in particle-associated and free-living alpha diversity between any two days or weeks were often as great or greater than between any two months across the sampling period. A potential explanation for this pattern may be that temporal variability between days in microbiome richness was obscured by more fine-scale temporal and spatial heterogeneity. Although the LMR is turbulent and generally well mixed, because of its high energy and complex currents (that may include gyres, eddies, and upwelling) patchiness is possible at local and sub-daily scales.

However, while differences in bacterial OTU richness were not predictable based on time interval of sampling for either component of the river microbiome, richness of OTUs was always greater in the particle-associated fraction compared to free-living assemblages. This observation is consistent with those made previously along the length of the Mississippi in mid-summer 2013 [17], highlighting that suspended particles are important microhabitat "hotspots" for bacterial production [18, 43], organic matter transformations [43, 44], and species richness in large river systems.

In contrast to temporal patterns in alpha diversity, we found that beta diversity of both particle-associated and free-living assemblages varied least on a daily sampling basis, more on a weekly basis, and most between samples separated by monthly intervals. At longer timescales, bacterioplankton assemblages separated by roughly six months were the most distinct from each other in composition, while those separated by more than six months up to a year gradually converged towards similarity. This parabolic pattern of community assemblage differences aligns partly with temperature being an important driver of community assembly in the LMR and other temperate aquatic environments [3–7]. However, in addition to temperature, shifts in composition were related to variability in dissolved N (highest in spring) and chlorophyll *a* (highest in late summer), indicating that fluctuations in nutrients contribute to seasonality of the river microbiome, and suggesting that the composition of bacterioplankton assemblages of

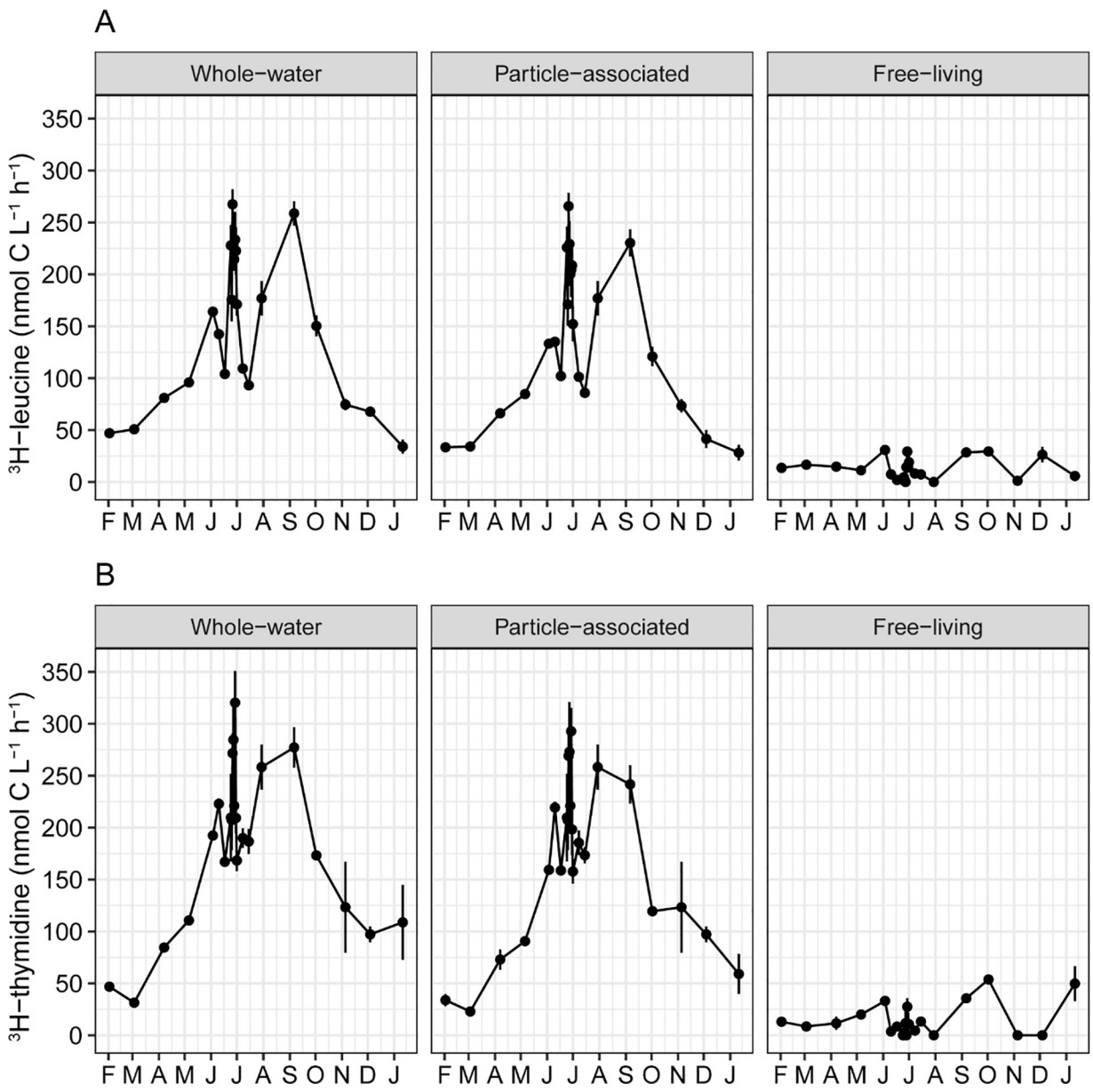

**Fig 8. Rates of bacterial production measured from whole-water, and from particle-associated and free-living cells between February 2013 and January 2014 using a [3]H-leucine and b [3]H-thymidine.** Rates of production are presented as means (± SE) for each date, n = 2–3.

such large rivers [7–10] may be predictable depending on the interaction of the temperature and nutrient regimes.

Patterns in composition were associated with changes in the relative abundances of bacterial taxa important in other large river systems [7–10, 16, 17, 45–50]. The principal environmental correlate of change in proportion of most phyla in particle-associated and free-living assemblages was temperature, to which Acidobacteria and Planctomycetes responded positively, and Bacteroidetes and Verrucomicrobia responded negatively. Taxa identified as Actinobacteria responded positively to low river flow, and contributed to substantial differences in the free-living microbiome between mid-summer and fall. Actinobacteria were observed previously during mid-July in 2012 in major tributaries of the Mississippi [16], and during mid-

July in 2013 along a 1,300 stretch of the Mississippi itself [17], to be in much higher proportions in free-living assemblages than in the particle-associated microbiome. These studies indicate that during low flow conditions aquatic members of Actinobacteria (e.g. order Actinomycetales) are consistently prominent within free-living assemblages. These taxa may be more competitive when discharge is low due to a reduction in the immigration of allochthonous bacteria from terrestrial sources [51], and/or as a consequence of increased time in transit [9, 47–50].

Differences in beta diversity of assemblages were maximized at around 180 days apart in sampling, regardless of the times of year being compared, while differences in bacterial alpha diversity did not vary with time interval. This is likely because microbiome composition varied along seasonal transitions in temperature as well as dissolved N and chlorophyll *a* concentrations, while bacterial richness oscillated unpredictably at short timescales. Bacterial production, in contrast, while ranging the most between cold and warm months, was nearly identical in spring and fall, indicating the dominant influence of water temperature on microbial metabolic activity. However, this was the case only for particle-attached assemblages, as production of free-living cells did not vary with changes in the environment. These results suggest that bacterial diversity and production in the LMR respond to different sets of drivers, resulting in different patterns of variation both within the river microbiome and across time.

## Conclusions

In this study, we found that variation in microbiome richness was unrelated to the timescale of change in the river environment, suggesting there is a high degree of local spatial variation in richness at any given moment in time. In contrast, variation in microbiome composition, as well as particle-associated production, was clearly related to temporal changes in the river environment. While production was driven almost exclusively by water temperature, the parabolic pattern of variation in dissimilarity indicates that composition was driven by changes in temperature interacting with temporal variation in other environmental factors having a strong seasonal pattern such as dissolved N and chlorophyll *a* concentrations. Our results indicate that temporal variability in composition of the LMR microbiome is not random; rather, there is successional change over monthly to seasonal timescales, with gradual divergence up to 180 days, followed by gradual reassembly thereafter up to at least 360 days distance in time.

## Acknowledgments

Field sampling assistance was provided by Derrick Bussan and Bram Stone. Laboratory assistance was provided by Tricia Lipson and Dr. Beth Baker. We thank Jason Hoeksema, Chaz Hyseni, and Jarrod Sackreiter for consultation on statistics.

## Author Contributions

**Conceptualization:** Colin R. Jackson, Clifford A. Ochs.

**Data curation:** Jason T. Payne.

**Formal analysis:** Jason T. Payne, Colin R. Jackson.

**Funding acquisition:** Colin R. Jackson, Clifford A. Ochs.

**Investigation:** Jason T. Payne, Colin R. Jackson, Justin J. Millar, Clifford A. Ochs.

**Methodology:** Jason T. Payne, Colin R. Jackson, Justin J. Millar, Clifford A. Ochs.

**Project administration:** Colin R. Jackson, Clifford A. Ochs.

**Resources:** Colin R. Jackson, Clifford A. Ochs.

**Supervision:** Colin R. Jackson, Clifford A. Ochs.

**Validation:** Jason T. Payne, Colin R. Jackson, Clifford A. Ochs.

**Visualization:** Jason T. Payne.

**Writing – original draft:** Jason T. Payne, Clifford A. Ochs.

**Writing – review & editing:** Jason T. Payne, Colin R. Jackson, Clifford A. Ochs.

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
