## [Decision Letter · Decision Letter 0]

9 Jan 2020

PONE-D-19-32405

Timescales of variation in diversity and production of bacterioplankton assemblages in the Lower Mississippi River

PLOS ONE

Dear Dr. Ochs,

Thank you for submitting your manuscript to PLOS ONE. After careful consideration, we feel that it has merit but does not fully meet PLOS ONE’s publication criteria as it currently stands. Therefore, we invite you to submit a revised version of the manuscript that addresses the points raised during the review process.

Both reviewers found the study to be well described and have suggested only minor revisions to better contextualize the study in the existing literature and provide clarifications to the methods/analysis.

We would appreciate receiving your revised manuscript by Feb 23 2020 11:59PM. To enhance the reproducibility of your results, we recommend that if applicable you deposit your laboratory protocols in protocols.io, where a protocol can be assigned its own identifier (DOI) such that it can be cited independently in the future. For instructions see: http://journals.plos.org/plosone/s/submission-guidelines#loc-laboratory-protocols

We look forward to receiving your revised manuscript.

Kind regards,

Christopher Staley, Ph.D.

Academic Editor

PLOS ONE

Journal Requirements:

"Field sampling assistance was provided by Derrick Bussan and Bram Stone. Laboratory

assistance was provided by Tricia Lipson and Dr. Beth Baker. We thank Jason Hoeksema, Chaz

Hyseni, and Jarrod Sackreiter for consultation on statistics. The UMMC Molecular and

Genomics Facility is supported, in part, by funds from the NIGMS, including Mississippi INBRE

(P20GM103476), Center for Psychiatric Neuroscience (CPN)-COBRE (P30GM103328),

 Obesity, Cardiorenal and Metabolic Diseases- COBRE (P20GM104357) and Mississippi Center

of Excellence in Perinatal Research (MS 465 -CEPR)-COBRE (P20GM121334). Funding for the

study was provided by NSF DEB 1049911."

'CAO

NSF DEB 1049911

National Science Foundation

https://www.nsf.gov/

The funders had no role in study design, data collection and analysis, decision to publish, or preparation of the manuscript.'

Please provide an amended Funding Statement that declares *all* the funding or sources of support received during this specific study (whether external or internal to your organization) as detailed online in our guide for authors at http://journals.plos.org/plosone/s/submit-now Please state what role the funders took in the study.  If any authors received a salary from any of your funders, please state which authors and which funder. If the funders had no role, please state: "The funders had no role in study design, data collection and analysis, decision to publish, or preparation of the manuscript."

Reviewers' comments:

Reviewer's Responses to Questions

**Comments to the Author**

1. Is the manuscript technically sound, and do the data support the conclusions?

Reviewer #1: Yes

Reviewer #2: Yes

2. Has the statistical analysis been performed appropriately and rigorously? 

Reviewer #1: Yes

Reviewer #2: Yes

3. Have the authors made all data underlying the findings in their manuscript fully available?

Reviewer #1: Yes

Reviewer #2: No

4. Is the manuscript presented in an intelligible fashion and written in standard English?

Reviewer #1: Yes

Reviewer #2: Yes

5. Review Comments to the Author

Reviewer #1: “Timescales of variation in diversity and production of bacterioplankton assemblages in the lower Mississippi River” seeks to describe patterns in particle-associated and free-living microbial assemblages at three different timescales using next generation sequencing combined with water physical and chemical paired measurements. The study is well organized and clearly written. Findings should be of interest to PLOS readers.

DNA extraction and sequencing quality controls. Authors performed DNA extractions followed by 16S rRNA gene amplification. Were any field blanks, method blanks, no template controls, or positive controls used during 16S rRNA gene amplification? Did authors include a method blank during sequencing? If so, please list controls and results in manuscript. If not, please mention that these controls were not included along with rationale.

Screen for covariate auto correlation. Authors do not report any correlation testing amongst covariates to identify potential confounding auto correlation. Please conduct correlation analysis among covariates used for statistical testing and report results. If some covariates end up being auto correlated, then repeat respective statistical tests with only covariates not auto correlated and revise manuscript as needed.

Minor Comments:

Line 112: Please include the maximum length of time samples remained frozen prior to testing. For example, (< xx months).

Line 127: Please include the maximum length of time samples remained frozen prior to testing. For example, (< xx months).

Line 174: Do you mean sampling “period” or “event”? I think you mean “event”, please clarify.

Line 249: Please provide range of high-quality sequence reads for individual sample sets.

Line 437: Please revise statement to, “This is likely because microbiome…”

Figure 1 caption: Please provide the USGS gage number in the caption description.

Reviewer #2: The manuscript by Payne et al describes temporal variability in bacterioplankton community structure and function in a river ecosystem. This research covers several important concepts that add valuable information to significant knowledge gaps in the literature including. Firstly, it links measurements of both structure and function, which is needed in more studies examining microbiome-ecosystem interactions. Secondly, it addresses temporal variability, which is not well understood in microbial ecology, especially at the microbiome scale. And thirdly, it addresses river ecosystems, which represent a dynamic and unidirectional flowing system that are quite different from more stable microbiome habitats that are commonly studied such as hosts, soils, and blue water marine systems. The manuscript is also well written and the results are presented in a manner that is broadly valuable beyond those interested in the Lower Mississippi River. Adding the distinction between particle-associate and free-living organisms is also an important contribution. I have the following suggestions to improve and clarify the manuscript prior to publication:

The introduction is well written and relevant. However the discussion of previous references seems a little thin. For example, lines 57-66 represent just speculation and hypothesis exploration on the part of the authors. I would prefer to see this space devoted to a little more detail about what was learned in previous studies related to these questions and what knowledge gaps still remain that are being addressed here.

There is also not really any discussion in the introduction related to “environmental change” and the factors (temp, chla, nutrients, etc.) that were measured in the study. What knowledge gap is being addressed here?

L 75-78: The hypotheses are also somewhat vague. What does “scale with time” mean? In relation to what type of environmental change that is being measured here? This is written as one hypothesis, but seems to actually be at least three.

L 100-102: I assume the more intensive sampling was conducted in the summer due to higher biomass/productivity/etc? It would be good to provide a brief rationale.

Are the sequence data being made publicly available?

The data analysis section is very well explained.

I suggest looking for opportunities to reduce wordiness in some of the results. E.g., L262: “more variability”; L273: “were more similar”; L288: “similarly variable in composition”

L280: I might be missing something, but it doesn’t seem like you can comment on something anything happening “on an annual basis” based on the patterns in a single year.

I’m not sure I understand the rationale for how the discharge data are used. Unless I’m missing something, those data are not used in the model selection. Why not? And if they aren’t used there, why include them?

The figures overall are very well done.

6. PLOS authors have the option to publish the peer review history of their article (what does this mean?). If published, this will include your full peer review and any attached files.

Reviewer #1: No

Reviewer #2: No

---

## [Author Response · Author response to Decision Letter 0]

2 Mar 2020

PONE-D-19-32405

Timescales of variation in diversity and production of bacterioplankton assemblages in the Lower Mississippi River

PLOS ONE

Review Comments to the Author

Reviewer #1: “Timescales of variation in diversity and production of bacterioplankton assemblages in the lower Mississippi River” seeks to describe patterns in particle-associated and free-living microbial assemblages at three different timescales using next generation sequencing combined with water physical and chemical paired measurements. The study is well organized and clearly written. Findings should be of interest to PLOS readers.

DNA extraction and sequencing quality controls. Authors performed DNA extractions followed by 16S rRNA gene amplification. Were any field blanks, method blanks, no template controls, or positive controls used during 16S rRNA gene amplification? Did authors include a method blank during sequencing? If so, please list controls and results in manuscript. If not, please mention that these controls were not included along with rationale.

Regarding the use of controls during 16S rRNA gene amplification, we have added the following explanation within the methods section:

Negative (no template) controls were used in all amplifications and consistently gave negative results. Such negative amplifications were also used as blanks in sequencing, yielding no sequence data. Positive controls were not needed as we have used these procedures successfully for a variety of samples types (12, 18, 19).

12. Payne JT, Millar JJ, Jackson CR, Ochs CA (2017) Patterns of variation in diversity of the Mississippi river microbiome over 1,300 kilometers. PLoS One 12: e0174890. doi: doi.org/10.1371/journal.pone.0174890

18. Shirur KP, Jackson CR, Goulet TL (2016) Lesion recovery and the bacterial microbiome in two Caribbean gorgonian corals. Mar Biol 163:238 doi:10.1007/s00227-016-3008-6

19. Weingarten EA, Atkinson CA, Jackson CR. (2019) The gut microbiome of freshwater Unionidae mussels is determined by host species and is selectively retained from filtered seston. PLoS One 14: e0224796. doi: doi.org/10.1371/journal.pone.0224796.

Screen for covariate auto correlation. Authors do not report any correlation testing amongst covariates to identify potential confounding auto correlation. Please conduct correlation analysis among covariates used for statistical testing and report results. If some covariates end up being auto correlated, then repeat respective statistical tests with only covariates not auto correlated and revise manuscript as needed.

As suggested, we conducted a cross-correlation matrix analysis of predictors. The table of correlation results is below. A priori, we assumed a correlation coefficient of 0.80 indicating auto-correlation. Based on this analysis, we do not think that statistical tests need to be repeated.

We added the following statement to the Methods section: “A cross-correlation matrix analysis of candidate predictors was performed. Predictors having a correlation coefficient ≥ 0.8 were not both included in the model for community composition.”

We think that this analysis and statement is sufficient to address potential confounding auto correlation among variables. We include the matrix table below for the sake of review, but we do not think it is necessary to add it to the manuscript.

 Temp TSS Chla C..mmol. N..mmol. P..mmol. Discharge

Temp 1.00 

TSS 0.06 1.00 

Chla 0.10 -0.20 1.00 

C..mmol. 0.60 0.23 -0.15 1.00 

N..mmol. 0.57 0.48 -0.49 0.57 1.00 

P..mmol. 0.63 0.17 -0.34 0.47 0.65 1.00 

Discharge 0.21 0.62 -0.52 0.42 0.70 0.33 1

Please note that for the revision we replaced nutrient ratios (C/N, N/P, C/P) as predictors with concentrations of individual nutrients (DOC, TDN, TDP). This change strengthens the predictive capability of the model, eliminates the redundancy of considering both C/N and N/P as predictors (because N varies much more than C or N), and has only a minor effect on multivariate model output. Temperature and N (replacing C/N) remain the most important predictors of particle-associated communities, and Temperature and Chla remain the most important predictors of free-living composition (Table 2). 

Minor Comments:

Line 112: Please include the maximum length of time samples remained frozen prior to testing. For example, (< xx months).

We kept samples frozen < 18 months prior to testing. Changes were made within the text.

Line 127: Please include the maximum length of time samples remained frozen prior to testing. For example, (< xx months).

We kept samples frozen < 18 months prior to testing. Changes were made within the text.

Line 174: Do you mean sampling “period” or “event”? I think you mean “event”, please clarify.

We meant sampling event, not period. Changes were made within the text.

Line 249: Please provide range of high-quality sequence reads for individual sample sets.

The range of high-quality sequence reads for individual sample sets is now provided within the manuscript.

Line 437: Please revise statement to, “This is likely because microbiome…”

We made this revision within the text.

Figure 1 caption: Please provide the USGS gage number in the caption description.

Gage height data was collected at a station run by the U.S. Army Corps of Engineers, not the USGS. We provided coordinates for this gage station (34°44'26.79" N 90°26'42.52" W) in the caption description for clarification.

Reviewer #2: The manuscript by Payne et al describes temporal variability in bacterioplankton community structure and function in a river ecosystem. This research covers several important concepts that add valuable information to significant knowledge gaps in the literature including. Firstly, it links measurements of both structure and function, which is needed in more studies examining microbiome-ecosystem interactions. Secondly, it addresses temporal variability, which is not well understood in microbial ecology, especially at the microbiome scale. And thirdly, it addresses river ecosystems, which represent a dynamic and unidirectional flowing system that are quite different from more stable microbiome habitats that are commonly studied such as hosts, soils, and blue water marine systems. The manuscript is also well written and the results are presented in a manner that is broadly valuable beyond those interested in the Lower Mississippi River. Adding the distinction between particle-associate and free-living organisms is also an important contribution. I have the following suggestions to improve and clarify the manuscript prior to publication:

The introduction is well written and relevant. However the discussion of previous references seems a little thin. For example, lines 57-66 represent just speculation and hypothesis exploration on the part of the authors. I would prefer to see this space devoted to a little more detail about what was learned in previous studies related to these questions and what knowledge gaps still remain that are being addressed here.

There is also not really any discussion in the introduction related to “environmental change” and the factors (temp, chla, nutrients, etc.) that were measured in the study. What knowledge gap is being addressed here?

We added a paragraph within the Introduction (second paragraph of Introduction) that discusses environmental factors that are characteristic of large river systems, and could vary at different timescales, and that were examined in this study.

L 75-78: The hypotheses are also somewhat vague. What does “scale with time” mean? In relation to what type of environmental change that is being measured here? This is written as one hypothesis, but seems to actually be at least three.

We replaced “scale with time” with “change more over longer timescales” to clarify our hypothesis. We also explicitly stated environmental factors that were measured during this study.

L 100-102: I assume the more intensive sampling was conducted in the summer due to higher biomass/productivity/etc? It would be good to provide a brief rationale.

We added the following as an explanation for intensive sampling during summer: “We chose to sample frequently during summer because this is a period of high bacterial production (Ochs et al. 2010), and potentially a period in which a high degree of short-term temporal variation could be detected.”

Are the sequence data being made publicly available?

Yes, all sequences can be accessed in the NCBI SRA database under the BioProject ID PRJNA358603.

The data analysis section is very well explained.

Thank you!

I suggest looking for opportunities to reduce wordiness in some of the results. E.g., L262: “more variability”; L273: “were more similar”; L288: “similarly variable in composition”

Changes were made within the text to reduce wordiness.

L280: I might be missing something, but it doesn’t seem like you can comment on something anything happening “on an annual basis” based on the patterns in a single year.

“Reassembling on an annual basis” was removed from the text to reflect that patterns were observed within a single year.

I’m not sure I understand the rationale for how the discharge data are used. Unless I’m missing something, those data are not used in the model selection. Why not? And if they aren’t used there, why include them?

In the original manuscript, we included discharge in Figure 1 to display the dynamic hydrological nature of the river. Although we did not include these data in model selection, the reviewer is correct that discharge could be an important predictor. Therefore, we have revised our model selection procedure to include discharge.

The figures overall are very well done.

Thank you!

---

## [Editor Report · Decision Letter 1]

13 Mar 2020

Timescales of variation in diversity and production of bacterioplankton assemblages in the Lower Mississippi River

PONE-D-19-32405R1

Dear Dr. Ochs,

We are pleased to inform you that your manuscript has been judged scientifically suitable for publication and will be formally accepted for publication once it complies with all outstanding technical requirements.

With kind regards,

Christopher Staley, Ph.D.

Academic Editor

PLOS ONE
---

## [Editor Report · Acceptance letter]

18 Mar 2020

PONE-D-19-32405R1 

Timescales of variation in diversity and production of bacterioplankton assemblages in the Lower Mississippi River 

Dear Dr. Ochs:

I am pleased to inform you that your manuscript has been deemed suitable for publication in PLOS ONE. Congratulations! Your manuscript is now with our production department. 

With kind regards,

on behalf of

Dr. Christopher Staley 

Academic Editor

PLOS ONE